# Systemic Antibiotic Use in Acute Irreversible Pulpitis: Evaluating Clinical Practices and Molecular Insights

**DOI:** 10.3390/ijms25021357

**Published:** 2024-01-22

**Authors:** Shahnawaz Khijmatgar, Gionata Bellucci, Luca Creminelli, Giulia Margherita Tartaglia, Margherita Tumedei

**Affiliations:** 1Complex Structure of Surgical Maxillofacial and Odontostomatology, Fondazione IRCCS Ca’ Granda, Ospedale Maggiore Policlinico, 20122 Milan, Italy; gionata.bellucci@policlinico.mi.it (G.B.); luca.creminelli@policlinico.mi.it (L.C.); 2School of Medicine, European University of Madrid, 28670 Madrid, Spain; giugitartaglia@gmail.com; 3Department of Biomedical, Surgical and Dental Sciences, University of Milan, 20122 Milan, Italy; margherita.tumedei@unimi.it

**Keywords:** pulpitis, acute pulpitis, irreversible pulpitis, chronic pulpitis, reversible pulpitis, antibiotics, systemic antibiotics

## Abstract

This scoping review systematically evaluates the use of systemic antibiotics in treating acute irreversible pulpitis, integrating clinical practice patterns with recent molecular insights. We analyzed clinical evidence on antibiotic prescription trends among dental professionals and examined molecular research advancements in relation to pulpitis. This review is intended to bridge the gap between clinical practice and molecular research, guiding more evidence-based approaches to treating acute irreversible pulpitis. Electronic databases were searched for relevant articles published in English based on the objective of the review. A second search using all identified keywords and index terms was undertaken across all the included databases. In addition, a reference list of identified articles was searched. Studies including original research, systematic reviews, meta-analyses, clinical trials, and observational and retrospective studies, all written in English and published from 2010 onwards, were included, and an analysis of the text words contained in the titles and abstracts of the retrieved papers and of the index terms used to describe the articles was performed. A total of N = 53 articles were selected. Altogether, N = 43 (76.79%) articles were cross-sectional studies, N = 4 (11.11%) were systematic reviews, and N = 3 (5.36%) were guidelines. The most frequent level of evidence was level VI (N = 43 (76.79%). The mean percentage of dentists who prescribed antibiotics to treat acute irreversible pulpitis was 23.89 ± 23.74% (range: 0.05–75.7). Similarly, for specialists, it was 22.41 ± 15.64 (range 2.2–50.4), and the percentage for undergraduates was 17.52 ± 20.59 (range 0–62.6). The significant developments in research models for pulpitis research and the characterisation of biomarkers have led to better management strategies. Concurrently, significant advancements in molecular research provide new understandings of pulpitis, suggesting alternative therapeutic approaches. Although there are guidelines available, increased rates of antibiotic prescription are still prevalent around the globe.

## 1. Introduction

The most common sequela of dental caries is ‘Pulpitis’. This term is used to describe the inflammation of the pulp. According to the WHO, 60–90% of school children and 98.9% of adults have cavities, leading to pain and discomfort, and dentists come across this problem routinely in dental practice [1]. Pulpitis occurs due to various etiological factors or irritants that trigger the inflammatory response in the pulp, resulting in pain. One of the methods employed in emergency care to manage pain due to pulpitis is the prescription of antibiotics. This may provide a temporary solution, but the results are poor. This practice has increased the exploitation of antibiotic prescriptions in dentistry [2]. The prevalence values of pulpitis in India estimated in one study were 60.7%, 68.2%, and 43.8% in molars, premolars, and anterior teeth, respectively [3]. In the USA, a total of 403,149 hospital-based Emergency Dental (ED) visits were attributed primarily to pulp in 2006. A periapical abscess without sinus involvement was the most common diagnosis for ED visits (accounting for 79.97% of all ED visits) [4].

In this review, we seek to comprehensively examine the current landscape of systemic antibiotic usage in the management of pulpitis. By exploring the extent of the existing literature, evaluating the evidence for antibiotic efficacy, determining the impact of diverse etiological factors, and identifying recommended diagnostic and treatment approaches, we aim to contribute to a more informed and evidence-based decision-making process for the management of pulpitis.

Given the variability existing in the available literature and the controversial nature of systemic antibiotic use in dentistry, we hypothesize that this scoping review will reveal a diverse range of opinions and findings regarding the efficacy of systemic antibiotics in the treatment of pulpitis. Moreover, we anticipate that this review will highlight the influence of etiological factors on the management approach, suggesting that tailored strategies based on clinical presentation and thorough investigation are recommended for optimal diagnosis and treatment.


**Research Questions:**
What is the extent of the literature available on the use of systemic antibiotics for pulpitis management?What evidence supports or questions the efficacy of systemic antibiotics in treating pulpitis?How is the management of pulpitis influenced by different etiological factors, such as carious and non-carious sources?What approaches are suggested for the optimal diagnosis and treatment of pulpitis considering clinical presentation and appropriate investigation?


## 2. Materials and Methods

### 2.1. Search Strategy

The search strategy for this scoping review was designed to be as comprehensive as possible, considering time and resource limitations. A three-step approach was followed. A literature search was conducted across various electronic databases, including PubMed, MEDLINE, Cochrane Library, and EMBASE. Keywords used were “pulpitis”, “antibiotics”, and related terms. The following search strategy was used: (“irreversible pulpitis”) OR (pulpitis)) AND (antibiotic). This was followed by an analysis of titles, abstracts, and index terms in retrieved papers. Later, a search using all identified keywords and index terms across all selected databases was conducted. Lastly, the reference lists of identified reports and articles were searched for additional sources. The review includes sources in the English language due to a lack of skill in other languages. A single search strategy was used to search for all types of evidence sources simultaneously, enhancing sensitivity. As the review progressed, additional keywords and sources were identified and incorporated into the search strategy, ensuring transparency and auditability. Collaboration with a research librarian or information scientist was instrumental in designing and refining the search.

### 2.2. Inclusion and Exclusion Criteria

#### 2.2.1. Inclusion Criteria

The inclusion criteria were as follows: original research articles, systematic reviews, meta-analyses, clinical trials, observational studies, and retrospective studies published in English from the year 2010 to the present. The target population was human participants diagnosed with pulpitis. The included studies had to have investigated the use of systemic antibiotics as part of pulpitis management. The reported relevant outcome for original studies was the percentage of dental professionals prescribing antibiotics to treat irreversible pulpitis. Exclusion Criteria: Studies published before the year 2010, editorials, letters, conference abstracts, case reports, animal studies, in vitro studies, and studies involving participants without a confirmed pulpitis diagnosis were excluded. Additionally, studies not focusing on the use of systemic antibiotics for pulpitis management, those reporting irrelevant outcomes, and those published in languages other than English were excluded. The language restriction was applied due to a lack of required interpretation skills. Duplicate publications reporting on the same study, non-peer-reviewed sources, and studies with inadequate data, or an unclear methodology were also excluded from this scoping review.

#### 2.2.2. Study Selection

Two independent reviewers performed initial title and abstract screening to identify potentially relevant articles. A full-text evaluation followed, adhering to the predefined inclusion criteria. Discrepancies were resolved through discussion and consensus. Studies that aligned with the scope of this review were included for data extraction.

#### 2.2.3. Data Extraction

A standardized data extraction form was created and piloted to ensure consistency. The information extracted included study characteristics, methodologies, participant demographics, antibiotic interventions, outcomes, and conclusions regarding antibiotic efficacy. Data extraction was performed independently by two reviewers, with any disagreements being resolved through discussion.

#### 2.2.4. Data Analysis

A narrative synthesis approach was employed to analyze and summarize the extracted data. Themes related to the extent of the literature, evidence supporting or questioning antibiotic effectiveness, the influence of etiological factors on management, and recommended diagnostic and treatment approaches were identified and discussed.

### 2.3. Rating the Level of Evidence

The rating of evidence in this study was conducted based on a hierarchical approach derived from different types of published papers. This hierarchical approach was designed to assess the strength and reliability of the evidence. The highest level, Level I, was assigned to evidence derived from systematic reviews or meta-analyses that encompassed all relevant randomized controlled trials (RCTs). Level II evidence was drawn from well-designed RCTs, known for their robustness and minimal bias. Level III was assigned to evidence obtained from well-designed controlled trials without randomization, providing valuable data on interventions or outcomes. Level IV encompassed evidence from well-designed case–control and cohort studies, contributing to our understanding of causation and associations. Level V was associated with evidence drawn from systematic reviews of descriptive and qualitative studies, offering insights into complex phenomena. Level VI involved evidence from single descriptive or qualitative studies, serving as initial exploratory sources. Finally, Level VII was applied to evidence derived from the opinions of authorities and reports made by expert committees, recognizing the significance of expert judgment in shaping clinical understanding and decision making. This structured approach to rating evidence enabled a comprehensive evaluation of the quality and applicability of the sources included in this study [5].

## 3. Results

### 3.1. Levels of Evidence in the Literature

Level VI evidence was predominant, accounting for a substantial number of 43 articles, representing approximately 76.79% of the total. Level I evidence, characterized by the highest quality and reliability, was assigned to seven articles, making up 12.50% of the total. Lastly, Level VII evidence, considered to have the lowest level of reliability and quality, was observed in six articles, making up 10.71% of the total. These findings emphasize the prevalence of different types of evidence, highlighting the need for more high-quality research to strengthen the body of evidence in this field (Appendix A). There were no contributions from Level II, i.e., evidence obtained from well-designed RCTs, or Level III, that is, evidence obtained from well-designed controlled trials without randomization.

### 3.2. Demographics

The dataset (cross-sectional surveys) consists of 44 articles, and the total sample size from the dataset is 45,240 individuals. The average response rate across the studies was approximately 61.27%. There was a wide range in the number of responses, i.e., N = 11,616 responses from dentists, N = 4283 responses from specialists, and N = 893 responses from another group.

Notably, the response rates exhibit variability, with a mean response rate of approximately 61.27% and responses ranging from as low as 1.8% to 100% (Appendix A). The mean response rates from specific groups include 60.29% for dentists; N = 59.75% (95% CI 35.57% to 83.93%) for specialists; and N = 57.83% for others (95% CI 28.49% to 87.16%). There were eight studies written by undergraduate dental students.

### 3.3. Prescription Rates among Dental Professionals

The dental professionals were categorized into dentists (who have bachelor’s degrees in dentistry), specialists (who have specialist degrees and training), undergraduates (who did not qualify as dentists and are/were engaged in their initial training or final years of their programs), and others (possessing medical qualifications), with dentists representing the largest group (31 individuals). Dentists had a mean antibiotic prescription rate of approximately 24.6%, while specialists, a smaller group of nine individuals, exhibited a mean rate of 22.41%. The “Others” category, comprising various dental professionals (four individuals), demonstrated the highest mean antibiotic prescription rate, amounting to 41.77%. Lastly, the analysis included eight undergraduate dental students, with a mean antibiotic prescription rate of 17.52%. These findings underscore the diversity of antibiotic prescription practices among different dental professionals, with engagement levels varying across the sampled groups (Appendix A).

### 3.4. Literature Mapping

A total of N = 53 articles examining the prescription of antibiotics for treating irreversible pulpitis were selected (Figure 1). The United States (USA) contributed significantly, accounting for 10 studies, representing approximately 20.63% of the total. Spain followed closely with seven studies, making up 11.11% of the corpus. India was also a notable contributor, accounting for three studies, or 6.35%. Furthermore, Saudi Arabia also presented four studies, reflecting a similar percentage. Italy and Australia both offered four studies each, collectively constituting 12.7% of the analyzed research. The rest of the countries, each contributing one to three studies, collectively make up the remaining part of the dataset.

The journals with the highest representation were the *International Endodontic Journal* (12.96%), with seven articles, and *Antibiotics* (5.56%), with three articles. Most studies are classified as cross-sectional studies, accounting for 42 instances, corresponding to approximately 77.78% of the total. Systematic reviews also featured prominently, accounting for four articles (7.41%). Review articles followed this group, accounting for four studies, representing 6.35% of the corpus. Guidelines accounted for three (5.56%) articles. There were also two separate instances of in vitro studies, contributing 1.85% to the total (Table 1 and Appendix A).

### 3.5. Molecular Developments in the Area of Irreversible Pulpitis

The molecular research with respect to irreversible pulpitis has witnessed significant developments. Researchers have published studies on various aspects, including the microbiota within teeth affected by irreversible pulpitis, the roles of autophagy and ceRNA networks in this condition, the impact of preoperative treatments, gene expression analysis, and the potential of various markers like substance P and IL-8 regarding assessing inflammation. Additionally, the analyzed studies explore the molecular and genetic underpinnings of pulpitis, including via the identification of specific bacteria and immune-related regulatory networks. The research also investigated stem cell potential, cytokines, and the diagnostic utility of dentinal fluid biomarkers. These studies collectively aim to advance our understanding of the molecular mechanisms, diagnostics, and potential treatments regarding irreversible pulpitis.

**Table 1 ijms-25-01357-t001:** Characteristics of the included studies.

									Prescription Rates (%)			
Author	Year	Country	Type of Study	Journal	Level of Evidence	Sample Size	Response Rate (%)	Type/Settings	General Practitioner	Specialists	Others	Undergraduates
Figueiredo, A.C.M., 2023 [6]	2023	Brazil	Cross-sectional Study	*Pesquisa Brasileira em Odontopediatria e Clínica Integrada*	Level VI	749	95.5	Dentists and dental graduates	75.7	-	-	-
Segura-Egea, J.J. 2022 [7]	2022	Spain	Guidelines	*Endodontic Advances and Evidence* *-Based Clinical Guidelines*	Level I	-	-	-	-	-	-	-
Agwan, M.A. 2022 [8]	2022	Saudi Arabia	Cross-sectional Study	*Medical Sciences*	Level VI	792	100	Dental practitioners and Specialists	2	-	-	-
Khaloufi, O. 2022 [9]	2022	Morocco	Cross-sectional Study	*Saudi Endodontic Journal*	Level VI	220	55	Dentists	4.9	-	-	-
D’Ambrosio, F. 2022 [10]	2022	Italy	Cross-sectional Study	*Healthcare*	Level VI	655	58.32	Dentists	19.4	-	-	-
Dias, N.M. 2022 [11]	2022	Columbia	Cross-sectional Study	*Acta Odontologica*	Level VI	559	57.2	Dentists	57.7	20.1	38.9	
Abbott, P.V. 2022 [12]	2022	Australia	Review	*International Endodontic Journal*	Level VII	-	-	-	-	-	-	-
Ramnarain, P. 2022 [13]	2022	Africa	Cross-sectional Study	*Health*	Level VI	122	72.1	Medical and dental	0.05	-	32.9	-
Carlsen, D.B. 2021 [14]	2021	USA	Cross-sectional Study	*Infection* *Control* *and Hospital Epidemiology*	Level VI	45,240	-	Dental visits	20.5	-	-	-
Arıcan, B. 2021 [15]	2021	Turkey	Cross-sectional Study	*Australian Endodontic Journal*	Level VI	1113	-	Dental students	-	-	-	3.1
Darwish, M.A. 2021 [16]	2021	Sudan	Cross-sectional Study	*Research Journal of Pharmacy and Technology*	Level VI	142	-	Dental students	-	-	-	20
Domínguez-Domínguez, L. 2021 [17]	2021	Spain	Cross-sectional Study	*Antibiotics*	Level VI	200	95	Dentists	12	-	-	-
Alobaid, M.A. 2021 [18]	2021	Saudi Arabia	Cross-sectional Study	*Infectious Drug Resistance*	Level VI	120	73.3	Dental university and government hospital	5.3	-	-	8
Drobac, M. 2021 [19]	2021	Serbia	Cross-sectional Study	*Antibiotics*	Level VI	628	25.16	Dentists	2.5	-	-	-
Ibrahim, N.I. 2021 [20]	2021	Lebanon	Rapid review	*Research Results in Pharmacology*	Level VII	-	-	-	-	-	-	-
Di Giuseppe, G. 2021 [21]	2021	Italy	Cross-sectional Study	*Antibiotics*	Level VI	971	32	Clinical records of the patients	-	-	82	-
Licata, F. 2021 [22]	2021	Italy	Cross-sectional Study	*Antimicrobial Agents Chemotherpy*	Level VI	1250	52.6	Dentists	13	-	-	-
Dibaji, F. 2021 [3]	2021	Iran	Cross-sectional Study	*Fronteir dental*	Level VI	400	95.7	Dentists	48.5	-	-	-
Karobari, M.I. 2021 [23]	2021	India	Cross-sectional Study	*Biomed Research International*	Level VI	350	90	Dentists	26.7	-	-	-
Darkwah, T.O. 2021 [24]	2021	Ghana	Cross-sectional Study	*Pan African Medical Journal*	Level VI	184		Prescriptions	-	-	-	-
Munitić, M.Š. 2021 [25]	2021	Croatia	Cross-sectional Study	Acta stomatologica Croatica	Level VI	657	23.96	Dentists	0.2	-	-	-
Gemmell, A. 2020 [26]	2020	United Kingdom	Cross-sectional Study	*British Dental Journal*	Level VI	1341	60	Dentists	25	-	-	-
Abraham, S.B. 2020 [27]	2020	UAE	Cross-sectional Study	*PLoS ONE*	Level VI	250	70	Dental practitioners and Specialists	12.3	9.1	13.3	-
Hussein, H.H. 2020 [28]	2020	NA	Molecular research	*International Journal of Pharmaceutical Research*	Level VII	-	-	-	-	-	-	-
Baudet, A. 2020 [29]	2020	France	Cross-sectional Study	*European Journal of Clinical Microbiology and Infectious Diseases*	Level VI	41,800	1.8	Dentists	50	-	-	-
Vasudavan, S. 2019 [30]	2019	USA	Cross-sectional Study	*Paediatric dentistry*	Level VI	3434	20	Dentists	48	41	-	-
Agnihotry, A. 2019 [31]	2019	USA	Cross-sectional Study	*Brazilian Dental Journal*	Level VI	403		Dentists	39.3	-	-	-
Lockhart, P.B. et al., 2019 [32]	2019	USA	Guidelines	*Journal of American Dental Association*	Level I	-	-	-	-	-	-	-
Agnihotry, A. 2019 [33]	2019	USA	Systematic review	*Cochrane Database Systematic Review*	Level I	-	-	-	-	-	-	-
Tampi, M.P. et al., 2019 [34]	2019	USA	Systematic review	*Journal of Americal Dental Association*	Level I	-	-	-	-	-	-	-
Salvadori, M. 2019 [35]	2019	Italy	Cross-sectional Study	*International Endodontic Journal*	Level VI	399	76	Dental Students	-	-	-	5
Dana, R. 2019 [36]	2019	Canada	Cross-sectional Study	*Clinical Research*	Level VI	1012	20.2	Physicians	57.4	-	-	-
Al Masan, A.A. 2018 [37]	2018	United Kingdom	Cross-sectional Study	*International Endodontic Journal*	Level VI	131	60	Dental students and dental practices	9.4	-	-	0
Alonso-Ezpeleta, O. 2018 [38]	2018	Spain	Cross-sectional Study	*Journal of Clinical and Experimental Dentistry*	Level VI	67	91.2	Dentists in endodontic programs	-	11.9	-	-
Martín-Jiménez, M. 2018 [39]	2018	Spain	Cross-sectional Study	*International Endodontic Journal*	Level VI	175	93.7	Final-year dental students	-	-	-	29.3
Segura-Egea, J.J. 2018 [40]	2018	Spain	Guidelines	*International Endodontic Journal*	Level I	-	-	-	-	-	-	-
Maslamani, M. 2018 [41]	2018	Kuwait	Cross-sectional Study	*Medical Principles and Practice*	Level VI	227	83.7	Dental clinics	4.8	-	-	-
Bolfoni, M.R. 2018 [42]	2018	Brazil	Cross-sectional Study	*International Endodontic Journal*	Level VI	13,853	4.44	Endodontists	-	2.2	-	-
Germack, M. 2017 [43]	2017	USA	Cross-sectional Study	*Journal of Endodontics*	Level VI	666	22.86	Dentists	1.75	-	-	-
Gottlieb, M. 2017 [44]	2017	USA	Systematic review	*Annals of Emergency Medicine*	Level I	-	-	-	-	-	-	-
Segura-Egea, J.J. 2017 [45]	2017	Spain	Review	*International Endodontic Journal*	Level VII	-	-	-	-	-	-	-
Segura-Egea, J.J. 2017 [46]	2017	Spain	Review	*International Dental Journal*	Level VII	-	-	-	-	-	-	-
AlRahabi, M.K. 2017 [47]	2017	Saudi Arabia	Cross-sectional Study	*Saudi Medical Journal*	Level VI	75	80	Dentists	6.7	-	-	-
Silva, M. 2017 [48]	2017	Portugal	Cross-sectional Study	*Revista Portuguesa de Estomatologia, Medicina Dentária e Cirurgia Maxilofacial*	Level VI	135	70	95 dentists	15.8	-	-	-
Fadare, J.O. 2017 [49]	2017	Nigeria	Cross-sectional Study	*Acta odontologica Scandinavica*	Level VI	607		Prescriptions	8.5	-	-	-
Wasan, H. 2017 [50]	2017	India	Cross-sectional Study	*Journal of Natural Science, Biology, and Medicine*	Level VI	667	80.8	Dentists	71.6	50.4		62.6
Tanwir, F. 2015 [51]	2015	Pakistan	Cross-sectional Study	*Oral Health Preventive Dentistry*	Level VI	709	100	Dentists	21	-	-	-
Garg, A.K. 2014 [52]	2014	India	Cross-sectional Study	*Journal of Antimicrobial Chemotherpy*	Level VI	1600	34.5	Dentists	71.6	-	-	-
Kaptan, R.F. 2013 [53]	2013	Turkey	Cross-sectional Study	*Ther Clinical Risk Management*	Level VI	1400	43	Dentists	29	-	-	-
Al-Harthi, S.E. 2013 [54]	2013	Saudi Arabia	Cross-sectional Study	*Saudi Medical Journal*	Level VI	165	100	Dental school	-	18.8	-	12.2
Segura-Egea, J.J. 2010 [55]	2010	Spain	Cross-sectional Study	*International Dental Journal*	Level VI	127	64	Oral surgeons	-	31.5	-	-
Skucaite, N. 2010 [56]	2010	Lithuania	Cross-sectional Study	*Medicina*	Level VI	1532	53.8	Dentists	2	-	-	-
Yingling, N.M. 2002 [57]	2002	USA	Cross-sectional Study	*Journal of Endodontics*	Level VI	3274	50.1	Dentists	-	16.76	-	-

**Level I**—Evidence from a systematic review or meta-analysis of all relevant randomized controlled trials (RCTs); **Level II**—evidence obtained from well-designed RCTs; **Level III**—evidence obtained from well-designed controlled trials without randomization; **Level IV**—evidence from well-designed case–control and cohort studies; **Level V**—evidence from systematic reviews of descriptive and qualitative studies; **Level VI**—evidence from single descriptive or qualitative studies; **Level VII**—evidence from the opinions of authorities and/or reports made by expert committees [5].

## 4. Discussion

### 4.1. Importance of the Review

The effects of the most-prescribed antibiotics for treating irreversible pulpitis are largely limited, with insufficient proof to support their significance. The misunderstanding of the pathogenesis of the pulp may have led to the observed increase in antibiotic prescription for treating pulpal diseases. In a study carried out in the USA, it was reported that 16.7% of endodontists prescribed antibiotics for the treatment of irreversible pulpitis [57]. Another study organized in The Netherlands reported that only a small proportion, i.e., 4.3%, of dentists continued to advise the use of antibiotics for irreversible pulpitis [31]. Similarly, a multistage sampling study in India confirmed that 71.6% of dentists over-prescribed antibiotics, mainly for treating irreversible pulpitis and acute apical periodontitis [52]. The over-usage of antibiotics is highly likely to lead to the growth in resistant strains of micro-organisms. There is no convincing evidence proving that penicillin-like antibiotics relieve pain and sensitivity, but many dentists continue to prescribe antibiotics. Similar concerns are discussed in other published articles [58,59,60,61,62,63,64,65,66]. Therefore, this review helps to reveal the effectiveness of antibiotics in the management of pulpitis and identify the available evidence regarding the management of pulpitis.

### 4.2. Pathogenesis of Pulpitis

It is very important to understand how pulpitis occurs because such knowledge can aid the management of patients for better outcomes. Pulpitis may occur due to a microbial insult, a chemical insult, or traumatic or iatrogenic factors. Caries and periodontal diseases are microbial in nature, while crown/root fractures and injuries are traumatic. An iatrogenic factor involves marginal leakage, dental material toxicity, or trauma caused by dental procedures. Hence, the type of management employed differs depending on the type of case and cause. The dental pulp is securely protected by dentin, cementum, and enamel, providing strong mechanical support. But when the degradation of the outer enamel or cementum layer occurs, the connective tissue of the dental pulp is rendered vulnerable to the ingress of toxins due to the exposed dentinal tubules. This allows the noxious components of the oral cavity to enter the pulp and cause pulpitis [67].

The presence of bacteria in the pulp initiates an inflammatory reaction and results in pulpal necrosis. The endotoxins and bacterial waste products that are produced by proteolytic bacteria exit through the apical foramen and accumulate in the peri-apical region or apex of the tooth. Accordingly, the immune system is triggered, and defense cells will not be able to enter the root canal, accumulating and resulting in bone loss. This apical region is free from bacteria; bacteria are only present in articles from sinus formation, actinomyces, etc. [68].

### 4.3. Permeability of Dentine to Bacterial Toxins

Physiologically and anatomically, dentine is a complex mineralized tissue in the tooth. The dentinal tubules consist of nerves, vessels, and dentinal fluid [69]. Bacterial plaque accumulation leads to a microbial insult afflicting the dentinal tubules, and dentin does not act as an effective barrier against the diffusion of bacterial components, as shown in some research [70]. There is evidence that bacterial components are carried to the pulp through dentine, wherein an inflammatory process is induced. Some articles report that bacterial toxins penetrate over short distances, and initial reactions begin to occur through the initiation of some host defense mechanism present in dentinal fluids. This shows that the inflammatory process arises either due to bacterial toxins or exposed dentinal tubules and/or the activation of signal substances arising from dentinal fluids.

### 4.4. Koch’s Postulate

Robert Koch tried to identify the specific organisms that caused specific diseases. Hence, he conceived of criteria that later came to be known as Koch’s postulate. These criteria outline the following:The microbes present are associated with the disease and its causative lesion;Upon isolation from the contaminated site and later transferal, the microbes should be grown on culture media;The microbes should induce a similar disease when a pure culture of the organism is introduced into a heathy host;The microbes should be able to re-isolatable from an experimentally infected host.

Taxa information (concerning the group to which an organism belongs) gives an understanding regarding the disease process that causes pulpitis. It helps to provide the reason behind “When the disease becomes acute and why”. This will aid our understanding of the role of antimicrobials in preventing infection. The acute form of acute periodontitis or other opportunistic infections cannot be explained by Koch’s Postulate [71]. According to a study, Prevotella melaninogenica was noticed in all acute and pus and/or tenderness articles but not in chronic articles. Other organisms included Peptostreptococcus spp., Eubacterium spp., and Campylobacter sputorum. These organisms were cultivated from a sample obtained from an intact pulp chamber of traumatized teeth. It was found that the composition of the microbiota in the root canal drives the course of the disease. Some black-pigmented bacteroid species contain certain taxa that induce acute inflammation, and other taxa do not contain these. Enteroccocus species were found to relate to periapical health rather than a disease. The organisms found after root canal treatment were from extra-oral sources or from food rather than stemming from therapy-resistant entities.

### 4.5. Pulpitis

The carious process that results in pulpitis and endodontic infections is predominantly governed by anaerobic organisms, predominantly Gram-negative bacteria. As a result, inflammation of the pulp occurs, which ranges from minimal inflammation to marked inflammation [72]. Pulpal pathosis is diagnosed based on the progress of the disease, corresponding to reversible pulpitis, irreversible pulpitis (asymptomatic), irreversible pulpitis (symptomatic), and pulp necrosis [73]. When dental caries reaches the pulp, reversible pulpitis occurs, and it is usually associated with mild inflammation of the pulp and mild intermittent pain. Thermal changes, especially those induced by cold drinks, will elicit this pain.

### 4.6. Why Is There a Need for Antibiotics?

Antibiotics are usually prescribed as a strategy for preventing infection and post-operative complications and for prophylaxis. These are prescribed by general dental practitioners and oral maxillofacial surgeons, and they are sometimes prescribed at the request of the patient if the dentist has not prescribed them. A recent study confirmed that more than two-thirds of 120 patients who were included in the study responded that they expected to receive antibiotics after a routine tooth extraction, and 70% of this group indicated that they would request them if not prescribed. These findings were surprising because the patients included were educated, i.e., they had at least an initial college or college degree [44,74,75,76,77]. There is a myth among healthcare professionals and patients that antibiotics play an important role in the prevention of disease (Table 2). Instilling proper education and awareness among these two groups will help to eliminate these myths related to antibiotics prescription.

### 4.7. Myths about Antibiotics

The choice between bactericidal and bacteriostatic agents is contingent upon various factors. Bactericidal agents, which swiftly eliminate bacteria, are considered indispensable for patients with compromised immune defenses. This holds particular significance in severe infections or conditions like sepsis, wherein prompt bacterial elimination is imperative. Conversely, when a patient’s natural defenses are unimpaired, bacteriostatic agents, impeding bacterial growth without immediate destruction, are often deemed satisfactory. Moreover, post-antibiotic effects (PAEs), referring to the prolonged suppression of bacterial growth after antibiotic exposure, are more enduring and reliable when administering bacteriostatic agents such as erythromycin and clindamycin than when administering bactericidal agents like betalactamase. The clinical efficacy of bacteriostatic agents appears to be less dose-dependent, contributing to their consistent post-antibiotic effects compared to bactericidal agents. The assertion that bactericidal agents are superior is supported by their rapid action, potential to prevent resistance, effectiveness in critical infections, and synergy with the host’s immune response, especially in compromised immune states (Table 2).

In addition, the idea that bacterial infections require a “complete course” of antibiotic therapy is a prominent myth regarding the management of irreversible pulpitis. It is crucial to dispel the notion of a “complete course” of treatment, as the duration of antibiotic therapy is contingent upon the clinical improvement of the patient. Contrary to a common misconception, sustained antibiotic use beyond the point of clinical remission is not universally necessary to prevent “rebound” infections. Specifically in orofacial infections, the idea of rebound has been debunked, provided the infection’s source is effectively eliminated. Orofacial infections typically endure for a brief period, often ranging from two to seven days. For patients undergoing antibiotic therapy for orofacial infections, daily clinical assessments are imperative. Ceasing antibiotic therapy becomes appropriate when substantial clinical evidence indicates the restoration of the patient’s host defenses, signaling control over the infection and its resolution. Thus, the effectiveness of an antibiotic treatment is intricately linked to ongoing clinical evaluation rather than a predetermined course of medication.

### 4.8. When to Prescribe Antibiotics

Basically, antibiotics are prescribed when there is systemic involvement due to infection. Antimicrobials are also prescribed in the following instances:As an adjunct to the management of a acute or chronic infection;In the management of active disease, e.g., acute necrotizing ulcerative gingivitis;When drainage cannot be established during the treatment of an uncooperative patient who requires hospitalization and must be operated on under general anesthesia;For a patient that needs to be treated in a hospital environment due to comorbidities [58] (Table 3).

Before prescribing antimicrobials, a comprehensive case history should be acquired. Patients should be examined carefully, and any signs of systemic involvement, for example, fever, lymph node involvement (lymphadenopathy), and swelling, should be searched for. This helps to rule out if a patient can be managed in a private dental setting or needs to be referred to a hospital [58]. In the Indian context, in September 2015, the chief scientific advisor for the WHO Regional Director for South-East Asia in New Delhi confirmed that guidelines will be published on the usage of antibiotics to reduce the over-prescription and tackle antibiotic resistance. The panel advised hospitals and related facilities in the country to develop their own protocols as a best practice to tackle the problem [61].

The body’s defense mechanism plays an important role in preventing the spread of infection, except in articles of immuno-compromised patients. According to the literature, 60% of an infection is removed by the host’s own defense mechanisms if the underlying cause is removed. Antibiotics only help in maintaining the balance between host defense and invasive agents. The most important factor that indicates whether antibiotics should be prescribed is the need for antibiotics rather than which one to prescribe. Asymptomatic articles like apical periodontitis of pulpal origin and chronic apical abscesses of endodontic origin do not require antimicrobial therapy for healing. Proper root canal cleaning with effective irrigating solutions will resolve such issues. For articles like acute apical abscesses with spontaneous pain and swelling that is localized intra-orally, proper root cleaning and irrigation shaping of the canal will help solve the problem. If the case involves cellulitis or an acute apical abscess with systemic involvement, then debridement, surgical incision, and an aptly chosen antimicrobial should be considered [44,46,58,75].

The indications for antibiotic prescription in cases of acute pulpitis extend beyond molecular and clinical aspects, encompassing specific medical conditions where antimicrobial therapy is crucial. These conditions include patients with heart valve replacements, whether mechanical or biological, especially those who have undergone surgery due to microbial endocarditis. Additionally, individuals with congenital complex heart defects, surgically corrected heart defects within the initial 6 postoperative months, or residual findings after correction fall within the scope of antibiotic indication. Patients with Grade V renal insufficiency requiring dialysis, those who have undergone organ transplantation, and individuals with hip joint prostheses or total knee arthroplasties in the first two years after surgery also necessitate antibiotic consideration. Moreover, antibiotic prescription is warranted for individuals who have undergone radiotherapy and require treatment of the irradiated jaw area, those on high-risk bisphosphonates with intravenous administration over an extended period, and HIV patients with granulocyte counts below 500/μL.

### 4.9. Solution to the Problem

The possible solution to the problem is education. One method of education is to teach from errors rather than principles. Special consideration is taken when it comes to prescribing antibiotics to patients suffering from infective endocarditis (IE). Patients visiting a dental practice for their appointment very rarely have taken their antibiotics. It is good practice for a dentist to select a different class of antibiotics if the patient is already on antibiotics for endocarditis prophylaxis. If possible, one should delay a dental procedure until at least 10 days after the completion of a course of antibiotics. This will allow for the usual oral flora to be reestablished. If an individual receiving long-term parenteral antibiotic therapy for IE requires dental treatment, the treatment should be timed to occur 30 to 60 min after the parenteral antibiotic therapy has been delivered. If the dosage of an antibiotic is inadvertently not administered before the procedure, the dosage may be administered up to 2 h after the procedure. However, administration of the dosage after the procedure should be considered only when the patient has not received the pre-procedure dose. Individuals with permanent kidney dialysis shunts should be administered a course of prophylactic antibiotics using the same protocol applied for IE 33.

### 4.10. Antibiotic Resistance

Several studies [78,79,80,81,82,83] have been carried out to determine the prevalence of antimicrobial resistance in India. A recent study revealed that, generally, resistance was observed for nalidixic acid (79%), followed by Co trimoxazole (75%) and ampicillin (72%). Moderate susceptibility was seen with fluoroquinolones, and good susceptibility was seen with Imipenem (15%) and cephalosporins. Antibiotic resistance induced by antibiotic prophylaxis has been reported recently, and the factors causing these problems need to be considered. A recent meta-analysis confirmed that between 38.7% and 50.9% of pathogens causing surgical site infections and 26.8% of pathogens causing infections after chemotherapy are resistant to standard prophylactic antibiotics in the USA [84].

### 4.11. Management

The following is a list of actions to be taken in the case of an acute dento-alveolar infection [58]:Acquire comprehensive medical and dental histories;Rule out the presence of fever, malaise, fatigue, dizziness, or other disability;Measure the patient’s pulse and temperature (a normal temperature is 36.3–37 °C);Define the nature and extent of the swelling;Identify the cause of the infection.

During this process, determine whether the patient should be treated in a dental or hospital setting. This can be performed by checking for the following [58]:Signs of septicemia, lethargy, and tachycardia;Elevated temperature i.e., 39.5 °C;Spreading cellulitis;Difficulty in breathing, swallowing, or closing one’s eyes;Dehydration;Trismus associated with a dental infection;Failure to respond to previous treatment;An uncooperative patient.

### 4.12. In Case of Chronic Dento-Alveolar Infections [58]

Chronic dento-alveolar infections are long-standing infections in the root canal system that result in the induction of a peri-apical infection. This can arise in decayed or root-filled teeth. The infection presents as a minor localized abscess and, in some articles, occurs in the sinus and rarely requires antimicrobial therapy unless there are signs of systemic involvement (fever, lymphadenopathy, and swelling).

### 4.13. Clinical Approaches

Precise diagnosis along with localization of the afflicted tooth should be given priority. The required testing and history documentation should be performed to achieve this. To make sure that the afflicted tooth is correctly identified, it is essential to diagnose the patient’s symptoms. A precise treatment plan is devised once the initial radiographs are analyzed thoroughly to determine the anatomical complexity of the tooth.

The application of a restorative or temporary sedative dressing is usually performed to treat reversible pulpitis.

Root canal therapy or extraction can be performed to treat irreversible pulpitis. An antibiotic can be employed if necessary, depending on the severity of the infection and the type of causative bacteria.

### 4.14. Overview of the Literature on the Effectiveness of Antibiotics in Treating Irreversible Pulpitis

Endodontic emergencies, occurring before, during, or after treatment, result from diverse pulp and root canal conditions. In this review, we aim to outline these emergencies, emphasizing the need for timely and comprehensive management. The 3D principle—diagnosis, definitive dental treatment, and drugs—guides this process. Diagnosis, the cornerstone, requires understanding various emergency-causing conditions, aided by a comprehensive classification. Treatment varies per diagnosis and includes root canal re-treatment or conservative approaches. Drugs complement treatment, which is tailored to the corresponding diagnosis. Addressing inflammation and infection distinctions is crucial for achieving effective pain relief and symptom resolution (Abbott PV 2022) [12].

### 4.15. Regional Variations

#### 4.15.1. USA

Carlsen (2021) [14] found that while many treatments aligned with ADA guidelines, extended antibiotic courses were common, highlighting the need for guideline adherence. Vasudavan (2019) [30] noted low adherence, especially in cases of tooth pain and localized abscesses. Agnihotry (2019) [31] highlighted inappropriate antibiotic use for irreversible pulpitis, noting that better practices were adhered to by educated dentists. Lockhart et al. (2019) [32] reported limited benefits and potential harm regarding antibiotic use. Tampi et al. (2019) [34] observed mixed effects. Germack (2017) [43] indicated that patient expectations were driving unnecessary prescriptions. Other studies emphasized the insufficient evidence regarding the efficacy of using antibiotics in dental care (Gottlieb 2017, Hoskin 2016, Yingling 2002) [44,57]. These findings underscore the need for rational antibiotic use in dentistry.

#### 4.15.2. Spain

Segura-Egea, J.J. (2022; 2017) [7,46] emphasized the effects of antibiotic prophylaxis on patients with compromised immunity and specific conditions like infective endocarditis or prosthetic joint replacements. The author also highlighted the overprescription of antibiotics in endodontic infections and the need for improved prescription habits and education. Domínguez-Domínguez, L. (2021) [17] revealed that 44% of dentists prescribed antibiotics for symptomatic irreversible pulpitis, with up to 27% not following current guidelines, indicating a need for improved antibiotic prescription habits among Spanish general dentists. Alonso-Ezpeleta, O. (2018) [38] found that dentists with postgraduate training in endodontics exhibited better adherence to international guidelines for antibiotic use. Martín-Jiménez, M. (2018) [39] assessed dental students’ knowledge of antibiotic indications in endodontics. Segura-Egea, J.J. (2018) [46] published a position statement on the use of antibiotics in endodontics. Segura-Egea, J.J. (2010) [7] noted that while many members of the SECIB selected appropriate antibiotics, some still prescribed them inappropriately in the management of endodontic infections.

#### 4.15.3. Saudi Arabia

The findings from multiple cross-sectional studies conducted in Saudi Arabia suggest that there is a concern regarding the appropriate prescription of antibiotics by endodontists and general dental practitioners (GDPs). While there is general adherence to global guidelines, instances of inappropriate antibiotic prescriptions were noted, particularly in cases of irreversible pulpitis, necrotic pulps without systemic involvement, and sinus tract infections. This indicates the need to improve knowledge and awareness among dental practitioners regarding the judicious use of antibiotics to combat antibiotic abuse and antimicrobial resistance, constituting a pressing issue in dental treatment practice in Saudi Arabia.

#### 4.15.4. Italy

In the study by Di Giuseppe (2021) [21], a widespread practice of providing inappropriate antimicrobial prescriptions for prisoners was identified, indicating a need for diagnosis-specific monitoring and the implementation of prison-focused antimicrobial stewardship policies. Licata (2021) [22] emphasized the necessity of developing practical antibiotic prescription guidelines with clear indications and an easy-to-follow regimen. Salvadori’s (2019) [35] findings highlighted the imperative to enhance the knowledge of Italian students regarding antibiotics and their appropriate use in endodontics. D’Ambrosio’s (2022) [10] study demonstrated a consistent trend in Italy, like other countries, showcasing a high prevalence of antibiotic misuse and overuse among Italian dentists, who employed a variety of antibiotic management strategies.

#### 4.15.5. India

In the study by Karobari, M.I. (2021) [23], awareness among dentists about antimicrobial prescription guidelines was found to be incomplete, indicating a requirement for further training and education to enhance evidence-based decision making to achieve improved practices and outcomes. Wasan H’s investigation in 2017 revealed a pattern of frequent irrational prescription of antimicrobials for odontogenic conditions, emphasizing the immediate and sustained need for guidelines and educational intervention programs in dentistry. This approach is crucial for enhancing the quality of antimicrobial prescribing practices within the dental field. Garg AK’s 2014 [52] study highlighted the issue of overprescription among oral healthcare providers in India, signifying a significant contribution to the global problem of antimicrobial resistance. The findings underscored the urgent necessity of raising awareness, both among the public and professionals, about the risks associated with antibiotic use.

The literature highlights significant regional variations in antibiotic prescription practices. These variations are influenced by factors such as local guidelines, dental education levels, and regional healthcare policies. For instance, dentists with postgraduate training in endodontics showed better adherence to international guidelines, suggesting that advanced education positively impacts prescription practices.

### 4.16. Dental Practitioner Experience

The articles analyzed encompassed 44 articles. The responses available for analysis related to the number of prescriptions, ranging from 67 (minimum) to 45,240 (Maximum). The response rates varied, with an average of 61%, indicating diverse participation levels among professionals. Dentists had a mean antibiotic prescription rate of 23.76%, while specialists, others, and undergraduates showed rates of 22.42%, 41.78%, and 17.53%, respectively. Appendix A provides a snapshot of antibiotic prescription patterns, highlighting variations in practices among different categories of dental professionals.

### 4.17. Role of Education and Awareness

The findings point to a critical need for enhanced education and awareness among dental professionals. There is a clear gap in understanding the appropriate use of antibiotics in dentistry, particularly in the treatment of irreversible pulpitis. This gap extends to the understanding of molecular advancements in pulpitis research, suggesting the need for alternative therapeutic approaches that could reduce reliance on antibiotics.

### 4.18. Individual Study Findings

The study by Abraham, S.B. (2020) [27] delves into the practices of antibiotic prescription in the context of endodontic infections in the UAE, showcasing a survey involving 174 respondents with a response rate of 70%. This research underscores the importance of responsible antibiotic use, as indiscriminate prescription can lead to the emergence of antibiotic-resistant microbes. It reveals the preferences of dental practitioners, with amoxicillin and erythromycin being popular choices, and indicates discrepancies between general dental practitioners and specialists. Additionally, the article identifies instances where antibiotics were prescribed incorrectly, notably in articles of irreversible pulpitis, necrotic pulps lacking systemic implications, and sinus tracts [27].

Meanwhile, Agnihotry A’s studies from 2014, and 2019 address the use of systemic antibiotics for treating irreversible pulpitis and the associated concerns about antibiotic resistance. The 2019 study, rated as possessing low overall evidence quality, investigated pain relief outcomes between antibiotic and placebo groups [31], while the 2019 study underscored the inappropriate prescription of antibiotics for irreversible pulpitis, highlighting the knowledge gaps among dentists in this regard [33]. These findings emphasize the need for responsible antibiotic prescription and more research to clarify the role of antibiotics in endodontic emergencies, especially irreversible pulpitis [85].

In Agwan MA’s (2022) study, it was observed that 2% of the participants reported that they would prescribe antibiotics for irreversible pulpitis, a condition characterized by inflammation of the dental pulp. However, this practice deviates from clinical guidelines, as antibiotics are generally not recommended for pulpitis articles, given that the condition is primarily related to inflammation rather than bacterial infection. Instead, the primary approach for managing pulpitis should involve proper endodontic treatment, such as root canal therapy, which addresses the root cause of the issue [8,37].

Al Masan AA’s (2018) study results indicate a significant focus on antibiotic prescription for conditions such as systemic complications (78%), acute apical abscesses (72%), and symptomatic apical periodontitis (28%). This study further highlighted variations in prescription practices between the Group G1 and Group G2 participants, with differences noted in various clinical scenarios. It is noteworthy that final-year undergraduate students seemed to be generally aware of the antibiotic resistance crisis, albeit with some gaps in their knowledge of antibiotic use guidelines for endodontic conditions. In contrast, general dentists displayed less awareness of antibiotic guidelines and sometimes deviated from them in their responses to clinical scenarios [37].

Furthermore, the short communication regarding the antibiotic prescription practices of dentists in Saudi Arabia [54] written by Sameer, E.A.H. (2013) found that a significant percentage of dentists prescribed antibiotics for endodontic conditions that typically do not require antimicrobial treatment. The rates of antibiotic prescription varied for different conditions, with some articles, like necrotic pulp with acute apical periodontitis and swelling, aligning with guidelines, while in others, there were deviations from recommended practices. This finding underscores the importance of educational initiatives for promoting rational antibiotic use in dental practice and combatting antibiotic resistance [18,38,47,86].

In a comprehensive analysis of the antibiotic prescription patterns of dental students, Arıcan, B. (2021) [15] collected data from 17 public and 3 private dental schools, accounting for 1113 final-year dental students. These students exhibited varying prescription behaviors across different clinical scenarios. Notably, 89.9% of the students prescribed antibiotics for acute apical abscess (AAA) articles with diffuse swelling, whereas 47.2% did so for AAA with localized swelling. Regional and university-type differences were evident in these patterns, with certain articles displaying significant variations. The students also exhibited diversity in their choice of antibiotic usage duration, with 41.7% opting for a 5–7-day period and 36.2% preferring to complete the entire course. Amoxicillin, co-amoxiclav, and clindamycin were favored antibiotics for patients without allergies, and prophylactic antibiotic use varied depending on the clinical condition. Awareness of antibiotic usage for post-endodontic scenarios showed some discrepancies, with students being less inclined to prescribe antibiotics for situations like irreversible pulpitis. These findings underscore the need for consistent guidelines and education in antibiotic prescription.

Baudet (2020) [29] conducted a survey involving 775 dentists wherein 455 complete questionnaires were included in the analysis. The dentists predominantly worked as general dental practitioners (81.5%) in self-employed roles (77.0%) within urban areas (53.8%). They reported conducting an average of 47 scheduled consultations, 10 emergency consultations, and eight antibiotic prescriptions per week. While around 75.3% claimed to possess knowledge of national recommendations, only 32.2% specifically mentioned the French guidelines stipulated by the National Agency for medicines (ANSM). The primary reasons for prescribing antibiotics were abscesses, cervicofacial cellulitis, and pericoronitis. Amoxicillin was the most commonly prescribed antibiotic, often in the form of 1 g administered b.i.d. for 6 or 7 days. This study provides insights into antibiotic prescription patterns and awareness among practicing dentists [59].

In a study by Bolfoni, M.R. (2018) [42], 13,853 questionnaires were distributed, with 615 being completed, resulting in a response rate of 4.44%. The respondents had a diverse demographic profile. In articles of pulpitis, only 1.1% of the respondents prescribed antibiotics, while in situations involving irreversible pulpitis with acute apical periodontitis, 6.2% prescribed antibiotics. This study provides insights into antibiotic prescription habits among dental professionals.

Furthermore, guidelines such as those recommended by the ADA advise against the prescription of antibiotics in articles of acute pulpitis, emphasizing the importance of prudent antibiotic use in dental practice (Carlsen, D.B. 2021) [14]. A study conducted by Daher, A. (2015) reaffirmed the significance of appropriate treatment, as antibiotic-based pulpotomies were associated with a lower survival rate compared to calcium hydroxide treatment [87]. Finally, D’Ambrosio, F. (2022) [10] conducted an online survey among Italian dentists to gauge their attitudes toward antibiotic prescription and awareness of antimicrobial resistance [63]. This study revealed that the primary reasons for antibiotic prescriptions included abscesses, extractions, and pulpitis. Despite their high awareness (98.9%) of antimicrobial resistance, only a minority (7.4%) consulted guidelines for antibiotic prescriptions. These findings underscore the importance of enhancing awareness of and adherence to antibiotic prescription guidelines among dental professionals to combat antimicrobial resistance effectively.

The study by Dahake, P.T. (2023) examined the prevalence of isolated bacterial species in the context of root canals. In this research, 50 teeth were examined, all of which contained both aerobic and anaerobic microorganisms [88]. This study revealed the presence of various bacterial species, including aerobic, microaerophilic, facultatively anaerobic, and obligate anaerobic bacteria, as well as specific species such as *Candida albicans* (*C. albicans*), *Bacillus subtilis* (*B. subtilis*), *Pseudomonas aeruginosa* (*P. aeruginosa*), *Staphylococcus aureus* (*S. aureus*), *Streptococcus mutans*, *Streptococcus mitis*, and others. The antibiotic resistance profiles of these bacteria were assessed, highlighting varying sensitivities and resistances to antibiotics like clindamycin, metronidazole, and doxycycline. This comprehensive study provides insights into the diverse bacterial compositions within root canals and their antibiotic resistance profiles.

The research conducted by Daher, A. (2015) [87] involved the treatment of primary molars in children and included a sample of 35 participants aged 3.6 to 9.4 years. These children had a total of 53 primary molars treated, with some undergoing CTZ pulpotomy and others receiving calcium hydroxide pulpectomy. The study analyzed various aspects of treatment outcomes, follow-up periods, and success rates. Notably, 62.2% of the primary molars treated with CTZ pulpotomy were rated as unsuccessful in the first year after the intervention. Tooth extraction was required for treatment failure articles, and radiographic and clinical aspects contributed to the categorization of articles as failures. The overall mean survival time for all treated molars was 15.2 months. The study also examined treatment outcomes based on treatment group and previous pulp diagnosis, revealing lower survival rates for articles with a necrotic pulp initial diagnosis and those treated with the mixed antibiotic paste used in CTZ pulpotomy. This research provides valuable insights into the outcomes of different dental treatments among pediatric patients [87].

The study by Yu, J. (2020) [50,89] conducted in Guangzhou aimed to assess the rational use of drugs, particularly analgesics and antibiotics, by dentists and their communication with patients regarding these medications. This research found that dentists in Guangzhou frequently prescribed amoxicillin, with percentages varying depending on dental conditions. For instance, amoxicillin was prescribed for 25% of articles involving acute pulpitis and for 80.1% of articles of acute apical abscesses. Furthermore, metronidazole was the second most recommended antibiotic, especially for articles of diffuse swelling after treatment of acute apical abscesses, with 89.6% of practitioners choosing this antibiotic. This study provides insights into prescription trends in Guangzhou, shedding light on the patterns of antibiotic usage in dental management.

The study by Wasan, H. (2017) [50] investigated the impact of dental qualifications and practice settings on antimicrobial prescription practices among dental practitioners in Delhi and the National Capital Region (NCR) of India. Notably, it revealed that antimicrobial prescription for acute pulpitis was significantly higher among those pursuing postgraduate degrees (62.2%) compared to qualified specialists and dental graduates. This suggests variations in prescription practices based on qualifications, highlighting the importance of understanding the factors influencing antibiotic prescriptions among dental practitioners.

Vessal G’s (2011) [90] study in Shiraz, the Islamic Republic of Iran, assessed the knowledge and practices of dental practitioners regarding the therapeutic use of antibiotics for treating patients with dentoalveolar infections. The study revealed that 25% of the surveyed dental practitioners believed it was appropriate to use antibiotics to treat patients with acute pulpitis. This finding was in line with studies conducted in Yemen and Kuwait, which also reported similar percentages of dentists prescribing antibiotics for acute pulpitis. However, this percentage was lower (13%) in a study conducted in England, indicating variations in antibiotic prescription practices among different regions.

In Vasudavan S’s (2019) [30] study, in which 3434 surveys were distributed, with a response rate of 20%, the research focused on understanding antibiotic prescription patterns among different groups of dental practitioners in different experience brackets. For irreversible pulpitis, antibiotics were reported to have been prescribed in 45% of responses, with variations among different groups. Dentists with less than 10 years of experience (39%) prescribed antibiotics significantly less than those with 10 or more years of experience, highlighting the influence of experience on antibiotic prescription practices.

Skucaite N’s 2010 study aimed to characterize the antibiotic prescription patterns during root canal procedures as reported by Lithuanian general dental practitioners. Questionnaires were distributed to all 2850 registered Lithuanian dental practitioners, and responses from 1431 licensed general dental practitioners were analyzed. Approximately 2% of the practitioners prescribed antibiotics for symptomatic pulpitis [56].

Dana R’s (2019) study evaluated the knowledge and practices of Ontario physicians with respect to managing non-traumatic dental conditions, particularly antibiotic usage. With a 20.2% response rate from 1012 physicians, the study found that 57.4% of physicians prescribed antibiotics for irreversible pulpitis articles, with amoxicillin being the most prescribed antibiotic. This research provided insights into antibiotic prescription practices followed by physicians in Ontario when addressing dental conditions [36].

Several studies have examined the antibiotic prescription patterns for dental conditions, particularly pulpitis. Darkwah TO’s 2021 study, conducted at the Ghana Police Hospital, analyzed 184 patient prescriptions (corresponding to 286 antibiotics) but did not specify the exact percentage of antibiotics prescribed for irreversible pulpitis or acute pulpitis [24]. Darwish MA’s 2021 study focused on dental students attending the University of Gezira, revealing that 30% of antibiotics prescribed for root canal treatments (RCTs) at Wad Madani dental teaching hospital did not align with the recent ADA guidelines [16]. They also reported a lack of knowledge about antibiotic prescription guidelines among Sudanese dentists and dental students. Deniz-Sungur D’s 2020 study, involving 1007 Turkish dentists, demonstrated that up to 10% of the participants prescribed antibiotics for symptomatic irreversible pulpitis [91]. Di Giuseppe G’s 2021 study conducted in Italian prisons found that 85.7% of prisoners diagnosed with symptomatic irreversible pulpitis with or without symptomatic apical periodontitis were prescribed antibiotics [21]. Dias NM’s 2022 research conducted in Colombia reported that 43.7% of dentists prescribed antibiotics for irreversible pulpitis with symptomatic apical periodontitis and that 57.2% prescribed them for symptomatic acute apical periodontitis [11]. Finally, Dibaji F’s 2021 cross-sectional study involving 400 general dentists in Iran indicated that antibiotic prescription ranged from 48.5% for articles of painful irreversible pulpitis to 97.3% for articles of pulp necrosis with acute apical periodontitis and preoperative symptoms [3]. These studies provide valuable insights into antibiotic prescription practices for pulpitis in various regions and among different groups of dental practitioners.

Drobac M’s 2021 study conducted in Serbia involved 628 dentists with a 25.16% response rate, where 1.3% of the respondents indicated antibiotic reliance for symptomatic irreversible pulpitis [19]. In D’Ambrosio F’s 2022 study conducted in Italy, antibiotic prescription for pulpitis corresponded to a level of 14.1% [10]. Fadare JO’s 2017 study conducted in Nigeria analyzed 607 prescriptions, revealing that 8.5% of the treated patients received antibiotics for acute pulpitis [49].

Fedorowicz Z’s 2013 Cochrane review investigates the effectiveness and safety of oral antibiotics in treating severe toothaches caused by irreversible pulpitis, a condition resulting from nerve damage inside a tooth. The ‘standard of care’ involves the immediate removal of the affected pulp, but in some regions, antibiotics are still prescribed. This review, based on evidence available as of February 2019, includes one study with 40 participants who were administered either penicillin or a placebo in addition to painkillers. The findings indicate that antibiotics do not significantly reduce toothache caused by irreversible pulpitis, and there was no difference in painkiller use between groups. The study’s limited size and low certainty of evidence emphasize the need for more high-quality research on antibiotic use for treating this condition [31].

Garg AK’s 2014 study conducted in India found that 73.4% of dental practitioners preferred amoxicillin, with the majority prescribing antibiotics for irreversible pulpitis and acute apical periodontitis [52]. Gemmell A’s 2020 survey involving general dental practitioners showed that 25% frequently prescribed antibiotics for irreversible pulpitis [26].

Yingling NM’s 2000 survey of American Association of Endodontists members reported that 16.76% prescribed antibiotics for irreversible pulpitis, while only 3.47% and 13.29% prescribed antibiotics for specific articles of irreversible pulpitis [57]. Germack M’s 2017 survey of endodontists indicated that antibiotics were prescribed for articles of irreversible pulpitis with mild symptoms (1.75%) and moderate symptoms (6.41%) [43]. The findings from these studies highlight the variability of antibiotic prescription practices for pulpitis among dentists in different regions.

Gottlieb, M. 2017 aimed to review the best available evidence on the utility of antibiotics for treating dental pain without evidence of an overt infection. There is insufficient evidence with which to support the use or disuse of empiric antibiotics to prevent pain or reduce infection rates. Further data are required to provide definitive recommendations. However, the use of empiric antibiotics is not without risks, and this should be considered considering the current evidence. Additionally, it is important to provide pain control and conduct a close follow-up with a dentist for a pulpectomy [44].

Karobari, M.I. 2021 conducted a survey among dentists around three different regions of the world: 26.7% of dentists were found to prescribe antibiotics for pulpitis [23]. Khaloufi O 2022 aimed to evaluate the prescription attitudes of dental practitioners in Northern Morocco when treating pulpal and periapical pathologies. A total of 121 (55%) practitioners (63 females and 58 males) responded. The average age was 37 ± 0.4 years, with a minimum of 24 years and a maximum of 62 years. The distribution according to age group showed that more than 75% of practitioners were < 45 years old, 51 practitioners were between 25 and 35 years old, and 41 practitioners were between 36 and 45 years old [9]. Marra F 2016 confirmed the existence of over-prescription due to the slow implementation of guidelines [75]. Martín-Jiménez M’s 2018 Spanish study found that for articles of irreversible pulpitis, up to 63% of students would prescribe antibiotics [39]. Maslamani M’s 2018 study found that of the 227 participants surveyed, 190 (83.7%) did not prescribe antibiotics for patients complaining of severe pain. Of the participants, 199 (87.7%) never prescribed antibiotics for reversible pulpitis with a normal periapical area [41].

### 4.19. Discrepancy between Guidelines and Clinical Practice

The successful translation of clinical guidelines into practice requires a multifaceted approach that addresses not only the dissemination of guidelines but also the education and engagement of healthcare professionals, the fostering of a culture that embraces evidence-based practices, and effective communication with patients to manage expectations and build trust in the decision-making process.

Following clinical guidelines, especially when it comes to not prescribing antibiotics for irreversible pulpitis, can be challenging for healthcare providers. One major problem is that not all doctors follow the guidelines the same way. Some may not want to change how they usually treat patients, especially if a change contrasts with their typical practices.

Another issue is that it can be difficult for dentists to keep up with the latest information. New guidelines might not always make their way into regular practice because not everyone is aware of or educated about the changes in how they should be treating patients.

Patient expectations also play a role. Sometimes, patients believe they need antibiotics, even if the guidelines suggest otherwise. Doctors might feel pressured to prescribe antibiotics just to make their patients happy. This situation is exacerbated when there is insufficient communication between doctors and patients about why antibiotics might not be necessary.

Money and legal concerns can also affect decisions. Doctors might worry about getting in trouble or upsetting patients if they do not follow what is seen as normal, even if the guidelines stipulate that they should act differently based on the evidence. So, even if the guidelines recommend not using antibiotics to treat irreversible pulpitis, these various factors can make it challenging for doctors to follow such recommendations in real-life situations.

Generally, the literature underscores a significant discrepancy between clinical guidelines and actual practices in the management of irreversible pulpitis. While guidelines generally advise against the routine use of antibiotics to treat this condition, the data indicate a prevalent trend of over-prescription across various regions. This disparity raises concerns about the effectiveness of and rationale behind current treatment approaches, especially considering the risk of antibiotic resistance.

### 4.20. Autophagy

The interesting concept of autophagy has been researched with respect to irreversible pulpitis, and it is worthwhile to shed light on this concept. Autophagy actively maintains cellular homeostasis by contributing to cellular metabolism, innate immunity, and cell survival. There is a significant relationship between autophagy and inflammation in infections, cancer, metabolic disorders, and liver diseases. It has been suggested that autophagy correlates with pulpitis. Most researchers have pointed out a close connection between autophagy and oral diseases such as periodontitis and pulpitis. Ye Yung (2023) screened nine hub lncRNAs as candidate regulators based on ceRNA networks, thereby offering a new reference for the further exploration of the association between autophagy and irreversible pulpitis [92]. A similar line of research was reported by Qi, S. (2019) regarding the expression of proteins in pulpitis [93].

### 4.21. Protein Characterization in Irreversible Pulpitis

The molecular characterization of proteins in irreversible pulpitis includes MMP-12, MMP-9, RANTES, MIP-2, MCP-1, MMP-2, MMP-1, and P-Selectin, which exhibited correlations of ≥0.8 with the duration of pain caused by cold. These proteins also displayed relatively strong correlations (0.5–0.75) with the level of pain experienced at presentation [94].

The levels of NKA, SP, IL-8, and MMP-8 vary depending on the clinical situation. For example, when the pulp tissue of symptomatic-irreversible-pulpitis-affected teeth and GCF specimens were compared to healthy tooth pulp tissue and GCF specimens, it wa sobserved that the levels of NKA, SP, IL-8, and MMP-8 increased dramatically. NKA, SP, IL-8, and MMP-8 levels were found to be considerably lower in GCF samples from teeth with symptomatic irreversible pulpitis 1 week after the inflamed pulp was removed. Finally, SP, IL-8, and MMP-8 levels were shown to be higher in pulp tissue samples from patients with symptomatic irreversible pulpitis who scored higher on pain scales than those who scored lower on pain scales [95].

### 4.22. Gene Expression and Biomarkers in Pulpitis

A list of normalized differentially expressed (DE) genes was created in Liu, L’s (2021) study in order to analyze the molecular pathways of pulpitis and find possible biomarkers for diagnosis [96]. Antibiotics have a great influence on both the host’s and micro-organisms’ genetic characteristics (for example, regarding interactions between antibiotic drugs and resistance genetic mutations [97]). Biomarkers aid in the qualitative detection of antibiotic resistance genes [98].

## 5. Conclusions

Based on the referenced studies and surveys, it was observed that healthcare providers often prescribe antibiotics based on uncertain diagnoses and the expectations of patients regarding these drugs. While antibiotics can reduce the risk of infection, their chronic use has led to a rise in resistant bacterial strains. Therefore, it is crucial for dental clinicians to be educated and trained in using antibiotics as a supportive measure rather than as a replacement for pain relievers. Clear clinical and prescription guidelines, along with accurate diagnostic techniques, are necessary to ensure the effective use of antibiotics without jeopardizing patient health. Individual health institutions and organizations involved in healthcare delivery should establish their own consensus guidelines for antibiotic prescription. This approach will enable the practice of ‘precision dentistry’ in antibiotic prescription.


**Recommendations**


The management of irreversible pulpitis is currently facing a critical challenge, as actual clinical practices often deviate from established guidelines. This issue mainly arises from inadequate adherence to these guidelines, excessive dependence on antibiotics, and educational gaps among dental professionals. The data highlight the urgent need for a unified approach to realign clinical practices with contemporary research and guidelines, focusing on the following areas:Educational Efforts—Improving the training of dental professionals in understanding the development of pulpitis and the judicious use of antibiotics.Supporting Guideline Compliance—Supporting adherence to clinical guidelines through ongoing professional development and regulatory initiatives.Utilizing Molecular Research—Incorporating recent molecular research findings into clinical practice to provide more specific and effective treatments.Managing Patient Expectations—Instructing patients on the nature of dental conditions and proper medication usage to decrease antibiotic prescriptions driven by patient demand.

## Figures and Tables

**Figure 1 ijms-25-01357-f001:**
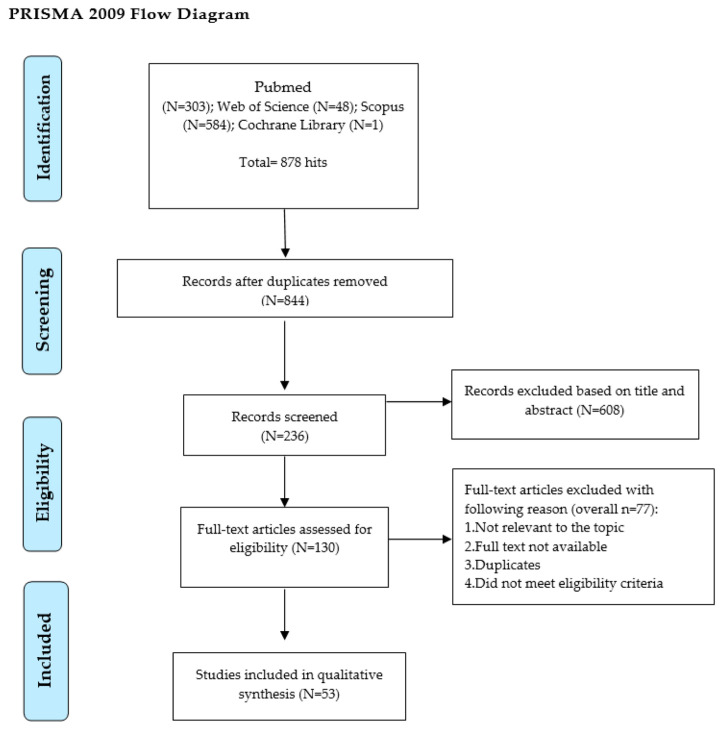
PRISMA Flow Chart.

**Table 2 ijms-25-01357-t002:** Myths about prescribing antibiotics [62].

1: Antibiotics cure patients. 2: Antibiotics are substitutes for surgical intervention. 3: The most important decision is which antibiotic to use. 4: Antibiotics increase the host’s defense against infection. 5: Multiple antibiotics are superior to a single antibiotic. 6: Bactericidal agents are always superior to bacteriostatic agents. 7: Antibiotic dosages, dosing intervals, and the duration of therapy are established for most infections. 8: Bacterial infections require a “complete course” of antibiotic therapy.

**Table 3 ijms-25-01357-t003:** Articles regarding the choice of whether to prescribe antibiotics [31,57,58,59,60,61].

Type of Case	Choice of Antibiotic	Dosage
Dental Caries	NA	NA
Acute Pulpitis	NA	NA
Asymptomatic Apical Periodontitis	NA	NA
Acute Apical Abscess Localised intra-orally	NA	NA
Chronic Apical Abscess	NA	NA
Acute Apical Abscess with systemic Involvement (Malaise, Swelling and Lymph Node Involvement)	First Choice Amoxicillin Second Choice Metronidazole Third Choice Clarithromycin	500 mg TID 400 mg TID 250 mg BID

## Data Availability

Not applicable.

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
