# Peer review of "Systemic Antibiotic Use in Acute Irreversible Pulpitis: Evaluating Clinical Practices and Molecular Insights"

_ijms, 2024, doi:10.3390/ijms25021357_

Round 1

Reviewer 1 Report

Comments and Suggestions for Authors

This is a scoping review about the use of antibiotics (ABs) for cases with irreversible pulpitis (IP) as regards clinical practice evidence and recent relevant molecular developments regarding infection and immunologic aspects of pulpitis. The review can be of benefit to patients, clinicians, and researchers. The review, however, has major comments that could affect its clarity and impact. Despite the effort spent in conducting such a review and the beneficial information included, yet the presentation of the review may be confusing rather than focusing. The review is very long to read with redundance at some parts and deficiency in others as well as insufficient coherence, consistency and clarity, thus, it needs to be more focused, specific, structured and summarized. The review gives higher emphasis on antibiotic prescription patterns but does not give the same attention to other relevant aspects/concepts e.g efficacy/effectiveness and safety of AB use with IP under a separate title, and the more sufficient coverage of other aspects like the microbiological profile in cases of irreversible pulpitis, which has several recent studies about it which were not included in this review (e.g. Arruda- Vasconcelos et al. (2020) “ investigation of microbial profile, levels of endotoxin and lipoteichoic acid in teeth with symptomatic irreversible pulpitis: a clinical study”). May be if fewer specific concepts/items are included, e.g. AB prescription pattern, with elaborate coverage for them would be better. Following a reporting guideline checklist, while writing the scoping review is very required e.g. PRISMA-ScR. The manuscript will benefit much from language revision regarding vocabulary, grammar, punctuation, and correct citation. Many sentences lack correct and/or sufficient references.

Title

-          The title needs to be more representative of the aim and be aligned with the conclusion(s) of the review.

Abstract

-          The abstract does not reflect much of what the text had to say and is focused mainly on prescription patterns. The aim was not consistently described throughout the manuscript with a kind of misalignment between the title and the different formats of the aim across the manuscript.  

-          “All types of studies were included” kindly specify and what were the exact clear eligibility criteria for studies to be included?!

-          The numbers of the studies will need revision as will be mentioned later.

-          Kindly mention how the level of evidence was evaluated. What was the scale used?!

Materials and methods

-          2.2.1. Inclusion criteria: it states that the included studies should investigate systemic AB use as part of pulpitis management. However, some of the included studies used local antibiotics as pastes after pulp extirpation!!!! E.g.  Daher et al. (2015) reference 57.

-          Table 1: how were the studies arranged in the table? Chronologically or alphabetically? Or if there is another system?

-          Three included studies are commentaries on 3 systematic reviews but they themselves are not systematic reviews. Kindly adjust throughout the manuscript. Zanjir et al. (2020) is a commentary on the Lokhart et al. (2019) systematic review. George et al. (2014) is a commentary on Fedorowicz et al. (2013) systematic review. Hoskin et al. (2016) is a commentary on Angihotry et al. (2016) systematic review).

-          The number of studies under each study design should be revised throughout the review. And the study design of each study should be correctly identified by a methodologist.

-           

-          Segura-Egea et al. (2016) is an editorial. Should it be included given that eligibility criteria exclude editorials in this review?!

Results

-          Kindly designate articles as “articles” throughout the review not “cases” or “samples” or “individuals” or any other term.  

-          “dental professionals are categorized into dentists, specialists, and others” kindly define and identify correctly and clearly the different categories of dental professionals.

-          3.4. Literature Mapping: Literature mapping is insufficient as it did not describe, categorize, and clearly present the different core concepts/key items/aims that these studies specifically assessed/studied. Studies on the guidelines for AB use were not clearly presented.  concepts not clearly categorized!!

Discussion

-          4.14. Studies in Literature: how were the papers arranged in this section? Was a chronological order followed?! The format of the authors’ names needs to be revised. were all the studies done by one author? The section needs much revision and summarizing collating studies under the same concept together!!! Kindly avoid repeating the same study at different locations.

-          4.15. Microbiota: kindly revise this section and include more relevant studies assessing the microbiological profile in cases with IP.

-          Reference numbers need revision throughout the manuscript. The format of authors’ names should also be revised throughout the review. E.g. reference 7 and 11 are the same. And there are other errors in referencing needing adjustment.

Comments on the Quality of English Language

The manuscript will benefit much from language revision regarding vocabulary, grammar, punctuation, and correct citation.

Author Response

Reviewer 1

This is a scoping review about the use of antibiotics (ABs) for cases with irreversible pulpitis (IP) as regards clinical practice evidence and recent relevant molecular developments regarding infection and immunologic aspects of pulpitis. The review can be of benefit to patients, clinicians, and researchers. The review, however, has major comments that could affect its clarity and impact. Despite the effort spent in conducting such a review and the beneficial information included, yet the presentation of the review may be confusing rather than focusing. The review is very long to read with redundance at some parts and deficiency in others as well as insufficient coherence, consistency, and clarity, thus, it needs to be more focused, specific, structured and summarized.

The review gives higher emphasis on antibiotic prescription patterns but does not give the same attention to other relevant aspects/concepts e.g efficacy/effectiveness and safety of AB use with IP under a separate title,

Thank you for the reviewers’ comments. We have added the following heading, and it is discussed and mentioned through different studies (Overview of literature on effectiveness of antibiotics in irreversible pulpitis).

and the more sufficient coverage of other aspects like the microbiological profile in cases of irreversible pulpitis, which has several recent studies about it which were not included in this review (e.g. Arruda- Vasconcelos et al. (2020) “investigation of microbial profile, levels of endotoxin and lipoteichoic acid in teeth with symptomatic irreversible pulpitis: a clinical study”).

Thank you for the reviewers’ comment and suggestion. This study aimed to analyze the microbial profile, endotoxin (LPS), and lipoteichoic acid (LTA) levels in infected dentine (ID) and root canals (RC) during various stages of root canal treatment in teeth with symptomatic irreversible pulpitis. Samples were collected before and after chemo-mechanical canal preparation (CMP) and after intracanal medication (ICM). Microbial analysis revealed a diverse bacterial population in ID and initial RC samples. LPS and LTA levels were higher in ID than initial RC samples, but canal preparation significantly reduced bacteria, LPS, and LTA levels. ICM had no additional effect on bacteria and LPS but reduced LTA levels.

May be if fewer specific concepts/items are included, e.g. AB prescription pattern, with elaborate coverage for them would be better. Following a reporting guideline checklist, while writing the scoping review is very required e.g. PRISMA-ScR. The manuscript will benefit much from language revision regarding vocabulary, grammar, punctuation, and correct citation. Many sentences lack correct and/or sufficient references.

Thank you for the reviewers’ comment and suggestion. We have used Grammarly for correcting the grammatical errors.

We have added these in the results section and also the literature we find the information (AB prescription pattern, with elaborate coverage for them would be better)

Title

-          The title needs to be more representative of the aim and be aligned with the conclusion(s) of the review.

Thank you for the reviewers suggestion. “Systemic Antibiotic Use in Acute Irreversible Pulpitis: Evaluating Clinical Practices and Molecular Insights”

We changed the title to above.

Abstract

-          The abstract does not reflect much of what the text had to say and is focused mainly on prescription patterns. The aim was not consistently described throughout the manuscript with a kind of misalignment between the title and the different formats of the aim across the manuscript.  

Thank you for the reviewers comments. We have done following changes

Background: This scoping review systematically evaluates the use of systemic antibiotics in acute irreversible pulpitis, integrating clinical practice patterns with recent molecular insights. We analyzed clinical evidence on antibiotic prescription trends among dental professionals and examined molecular research advancements in pulpitis. The review aims to bridge the gap between clinical practice and molecular research, guiding more evidence-based approaches to treating acute irreversible pulpitis.

Methods: Electronic databases were searched for relevant articles based on the objective of the review published in English. A second search using all identified keywords and index terms was undertaken across all included databases. In addition, the reference list of identified reports and articles were searched.  Studies including original research, systematic reviews, meta-analyses, clinical trials, observational and retrospective studies, all in English and published from 2010 onwards were included, and analysis of the text words contained in the title and abstract of retrieved papers, and of the index terms used to describe the articles was done. Results: A total N=54 articles were selected. N=43 (76.79%) were cross-sectional studies, N=4 (11.11%) systematic reviews and N=3 (5.36%) were guidelines. The most frequent level of evidence was at the level VI (N=43 (76.79%). The dentists mean percentage who prescribed antibiotics in acute irreversible pulpitis was 23.89±23.74% (range 0.05-75.7). Similarly, for specialists it was 22.41±15.64 (range 2.2-50.4) and undergraduates were 17.52±20.59 (range 0-62.6). A significant development in research models for pulpitis research, and characterisation of biomarkers has led to better management strategies. Concurrently, significant advancements in molecular research provide new understandings of pulpitis, suggesting alternative therapeutic approaches.

Conclusion: Although there are guidelines available, increased rates of antibiotic prescription are still prevalent around the globe.

-          “All types of studies were included” kindly specify and what were the exact clear eligibility criteria for studies to be included?! The numbers of the studies will need revision as will be mentioned later.

Thank you for the reviewer comments and we have included the following. Studies including original research, systematic reviews, meta-analyses, clinical trials, observational and retrospective studies, all in English and published from 2010 onwards were included,

-          Kindly mention how the level of evidence was evaluated. What was the scale used?!

Thank you for the reviewer comments and we have included the following. Please find the link https://libguides.mskcc.org/ebp/evidence

Materials and methods

-          2.2.1. Inclusion criteria: it states that the included studies should investigate systemic AB use as part of pulpitis management. However, some of the included studies used local antibiotics as pastes after pulp extirpation!!!! E.g.  Daher et al. (2015) reference 57.

Thank you for the reviewers’ comments. We deleted the articles and reference.

-          Table 1: how were the studies arranged in the table? Chronologically or alphabetically? Or if there is another system?

Thank you for the reviewers’ comments. We have arranged the table in the decreasing order of the year.

-          Three included studies are commentaries on 3 systematic reviews but they themselves are not systematic reviews. Kindly adjust throughout the manuscript. Zanjir et al. (2020) is a commentary on the Lokhart et al. (2019) systematic review. George et al. (2014) is a commentary on Fedorowicz et al. (2013) systematic review. Hoskin et al. (2016) is a commentary on Angihotry et al. (2016) systematic review).

Thank you for the reviewers’ comments. We have deleted the references and adjusted in the text

-          The number of studies under each study design should be revised throughout the review. And the study design of each study should be correctly identified by a methodologist.

 Thank you for the reviewers’ comments. We have deleted the references and adjusted in the text. The type of studies has been validated again.

-          Segura-Egea et al. (2016) is an editorial. Should it be included given that eligibility criteria exclude editorials in this review?!

 Thank you for the reviewers’ comments. We have deleted the reference and adjusted in the text

Results

-          Kindly designate articles as “articles” throughout the review not “cases” or “samples” or “individuals” or any other term.  

Thank you for the reviewers’ comments. We have deleted and adjusted in the text

-          “dental professionals are categorized into dentists, specialists, and others” kindly define and identify correctly and clearly the different categories of dental professionals.

Thank you for the reviewers comments. We have adjusted the following; The dental professionals are categorized into dentists (who have Bachelor’s degree in dentistry), specialists (have specialist degree and training), and others (Medicine qualifications),

-          3.4. Literature Mapping: Literature mapping is insufficient as it did not describe, categorize, and clearly present the different core concepts/key items/aims that these studies specifically assessed/studied. Studies on the guidelines for AB use were not clearly presented.  concepts not clearly categorized!!

Thank you for reviewers’ comments. The specific content of each study has been discussed in different sections. The literature mapping is highlighting the type of studies, data from various countries, with the United States, Spain, India, Saudi Arabia, Italy, and Australia being significant contributors. The studies are primarily cross-sectional, with a notable presence of systematic reviews, review articles, guidelines, cohort studies, molecular research, and commentaries. This section highlights the prevalence of cross-sectional studies, indicating a need for more high-quality research in this field. However, it seems that the categorization and clear presentation of core concepts, key items, or aims specifically assessed or studied in these articles could be more detailed, as well as a clearer presentation of guidelines for antibiotic use.

Top of Form

Discussion

-          4.14. Studies in Literature: how were the papers arranged in this section? Was a chronological order followed?!

Thank you for the reviewers’ comments, we referenced it based on the year

The format of the authors’ names needs to be revised. were all the studies done by one author? The section needs much revision and summarizing collating studies under the same concept together!!! Kindly avoid repeating the same study at different locations.

Thank you for the reviewers’ comments. The authors format mentioned was based on number of authors like for example, if the article is written by less than 6 authors, then only first author and year was mentioned. If the articles, is written by more than 6 authors then et.al is added.

We have corrected the repetition of the same study.

-          4.15. Microbiota: kindly revise this section and include more relevant studies assessing the microbiological profile in cases with IP.

Thank you for the reviewers’ comments.  

-          Reference numbers need revision throughout the manuscript. The format of authors’ names should also be revised throughout the review. E.g. reference 7 and 11 are the same. And there are other errors in referencing needing adjustment.

Thank you for the reviewers’ comments. We have adjusted the referencing and the order of studies.

Reviewer 2 Report

Comments and Suggestions for Authors

1.     Comprehensiveness and Relevance of Literature Review: The paper presents an extensive review of various studies conducted on antibiotic prescription patterns in dental practice across different regions and contexts. However, there is an opportunity to further streamline the literature review to enhance its focus and relevance to the core topic. Consider condensing similar studies and emphasizing findings that directly contribute to the narrative of antibiotic prescription trends and their implications.

2.     Clarity and Structure: The manuscript could benefit from a more structured presentation. The current format, while detailed, can be overwhelming and somewhat repetitive. Grouping similar studies and discussing them under subheadings could enhance readability and clarity. Subsections could include regional variations, differences based on dental practitioner experience, and variations in prescriptions for specific dental conditions.

3.     Methodological Analysis: The paper would be strengthened by a more critical analysis of the methodologies of the reviewed studies. This includes discussing the strengths, limitations, and potential biases in these studies. Such an analysis would add depth to your review, allowing readers to better understand the robustness of the conclusions drawn.

4.     Discussion and Conclusion: While the paper touches upon the discrepancy between clinical guidelines and actual practice, a more focused discussion on this aspect would be beneficial. Highlighting specific guidelines, how they contrast with current practices, and the reasons for these discrepancies would provide practical value to the paper. The conclusion section can be expanded to more explicitly address the implications of your findings for dental practice and policy. Discuss how these insights could inform the development of more effective guidelines, training programs, and policy interventions to optimize antibiotic use in dentistry.

5.     Figures and Tables: If possible, consider including summary tables or figures that synthesize the data from various studies. This could provide a quick visual reference for readers to understand key trends and comparisons.

6.       Language and Grammar: The manuscript is generally well-written but would benefit from a careful proofreading to correct minor grammatical errors and enhance sentence structure for improved readability.

Author Response

Thank you for the reviewers comments. Attached are the reply to the reviewer

Reviewer 3 Report

Comments and Suggestions for Authors

In their scoping review entitled „Acute Irreversible Pulpitis and Antibiotics: A Scoping Review of Clinical Practice Evidence and Overview on Recent Molecular development” Khijmatgar et al. systematically analysed the antibiotic prescription habits of dentists in cases of irreversible, painful pulpitis. The molecular part is weak, does not fit and must be deleted.

 General comments:

Title: Why this second part about “Overview on Recent Molecular development”. Is this truly justified or related to the journals title only.

 Materials and Methods:

The search strategy (important for reproducibility) is superficially reported. For instance, the sentence “Keywords used were "pulpitis," "antibiotics," and related terms.” Why not presenting the true search algorithm including Boolean Operators?

 Table 1: an important but also very lengthy table. The introduction of “Levels” is interesting but the source for such a classification (Nr. 88, which is a book-chapter)) not correctly cited. For instance there are three authors, what is the publisher etc.?

The order of this table could be optimized (ordered by level?) and more transparent. You might summarize the compressed definition of such levels in a legend also.

Figure 1 (PRISMA chart): please edit by style (blue boxes and flow-chart do not exactly fit) and contents, f.i. how do you come from n=395 to n=236?

Results

2.3 Rating of Evidence: you might add a comment about the levels (e.g. Levels II and III) NOT applied/found here.

3.2 Demographics. I do not understand –under this chapter title and without having a look to Suppl. Material- the sentence “The dataset consists of 44 samples, with a wide range in the number of responses, as reflected by a mean of 2967.909 and a standard deviation of 9203.735.” Please re-write.

3.3. Dental professionals: “dentists, specialists, and others”, the undergraduates are missing in this list but mentioned later. Include here as well.

3.5. This is a bad chapter according to my taste and only included because of the series title (IJMolecularSciences). It is a mish-mash of many things including cellular level (microbial, stem cells; which is still not molecular) and does not give any true review or information. The introductory sentence “The molecular research with respective irreversible pulpitis have significant developments.” reflects this problem quite well. Very artificial. Please allow to  delete this chapter (or make a second "molecular" review).

 Discussion:

4.2 Practices:

“A cross-sectional survey was conducted from January 2014 to February 2014”; the shortest survey I have ever heard of; please check data.

4.3-4.6: this makes this review again a very strange one. The etiology of acute pulpitis is a completely different topic. It belongs to “Introduction” but would be  inappropriate there too. Your article is not at the molecular basis of the disease but at the very end, to give an overview about the questionable impact of antibiotics. You are making here a balancing act to bring molecular background and practice evidence together. It would need a book to reach this goal. Please delete or summarize and move this to introduction. There is much more to discuss on the practical evidence level with so many articles included.  

4.7-4.12: these chapters are too universal, miss the point and exceed the length of most readers' ability to concentrate. We do not need an overview about antibiotics in general or in general dentistry. There is enough to discuss about the usage in pulpitis. However, I do like your Table 2. But it is entirely from a different source. Do you have the rights to present (repeat) this 1:1? Points 6 and 8 need discussion. It is only half a myth. Bactericide would of course be better than bacteriostatic activity but it is not an immanent feature of some drugs. “Complete course” needs also explanation; usually antibiotics should be given as prescribed (complete course) but we have single-shots also for prevention. 

4.14. this is the only part I would accept here as the appropriate discussing (plus 4.1 and 4.2). But I do not have the time to check details here, I have to admit.

4.15-4.18 this is rather bad, especially the microbiological part (this reviewer is professional in theoretical dental but also a microbiologist). It also does not fit to the clinical evidence. Please delete.

What is missing (after all these lengthy summaries of molecular and clinical aspects) are true indication for antibiotic usage in acute pulpitis: Heart valve replacement with mechanical or biological prostheses; Condition after microbial endocarditis; Congenital complex heart defects with cyanosis; Surgically corrected hearst defects within the first 6 postoperative months; Surgically corrected heart defects with residual findings; Grade V renal insufficiency (requiring dialysis); Condition after organ transplantation (for life); Patients with hip joint prostheses or total knee arthroplasties in the first 2 years after surgery; Condition after radiotherapy, if the treatment takes place in the irradiated jaw area bisphosphonates (high-risk category (intravenous administration over a longer period of time), and –finally - HIV patients with < 500 granulocytes/μl blood. A discussion here is only very rudimental given.

So please concentrate on the clinical evidence to reach at least one goal! Ask for a second review on the molecular part.

Comments on the Quality of English Language

no comments; language acceptable. 

Author Response

Thank you for the reviewers comments. Please find the attached reply to reviewers comments.

Round 2

Reviewer 1 Report

Comments and Suggestions for Authors

Thank you for the significant changes. There may still some minor adjustments needed on the format of writing author names within the body of the manuscript.

Comments on the Quality of English Language

The language has been improved.

Author Response

Thank you for the reviewers comments. We have corrected some of the author names as suggested.

Reviewer 2 Report

Comments and Suggestions for Authors

Please check the grammar and typos before the proof.

Author Response

Thank you for the reviewers comments. 

We have put efforts to make spelling and grammatical corrections using word tools “Review”.